# Clinical Application of Liquid Biopsy in Pancreatic Cancer: A Narrative Review

**DOI:** 10.3390/ijms25031640

**Published:** 2024-01-29

**Authors:** Elena Ramírez-Maldonado, Sandra López Gordo, Rui Pedro Major Branco, Mihai-Calin Pavel, Laia Estalella, Erik Llàcer-Millán, María Alejandra Guerrero, Estrella López-Gordo, Robert Memba, Rosa Jorba

**Affiliations:** 1HBP Unit, General Surgery Department, Joan XXIII University Hospital, 43005 Tarragona, Spain; mpavel.hj23.ics@gencat.cat (M.-C.P.); lestanela.tgn.ics@gencat.cat (L.E.); mariale.guerrero08@gmail.com (M.A.G.); rmembai.hj23.ics@gencat.cat (R.M.); rjorba.hj23.ics@gencat.cat (R.J.); 2Medicine and Surgery Department, Rovira i Virgili University, 43204 Reus, Spain; 3General Surgery Department, Maresme Health Consortium, 08304 Mataro, Spain; slopezgord@csdm.cat; 4General Surgery at Garcia de Orta’s Hospital, 2805-267 Almada, Portugal; 5Affinia Therapeutics Inc., Waltham, MA 02453, USA; elopezgordo@affiniatx.com

**Keywords:** pancreatic ductal adenocarcinoma, liquid biopsy, circulating tumor cells, circulating tumor DNA, circulating free tumor DNA, extracellular vesicles

## Abstract

Pancreatic ductal adenocarcinoma contributes significantly to global cancer-related deaths, featuring only a 10% survival rate over five years. The quest for novel tumor markers is critical to facilitate early diagnosis and tailor treatment strategies for this disease, which is key to improving patient outcomes. In pancreatic ductal adenocarcinoma, these markers have been demonstrated to play a crucial role in early identification, continuous monitoring, and prediction of its prognosis and have led to better patient outcomes. Nowadays, biopsy specimens serve to ascertain diagnosis and determine tumor type. However, liquid biopsies present distinct advantages over conventional biopsy techniques. They offer a noninvasive, easily administered procedure, delivering insights into the tumor’s status and facilitating real-time monitoring. Liquid biopsies encompass a variety of elements, such as circulating tumor cells, circulating tumor DNA, extracellular vesicles, microRNAs, circulating RNA, tumor platelets, and tumor endothelial cells. This review aims to provide an overview of the clinical applications of liquid biopsy as a technique in the management of pancreatic cancer.

## 1. Introduction

Pancreatic ductal adenocarcinoma (PDAC), ranking as the fourth leading cause of cancer-related deaths globally, presents a grim picture with a 5-year survival rate below 10%. This high mortality rate is due to late detection and notable resistance to chemotherapy [1,2,3,4,5,6,7,8].

Earlier diagnoses and more effective application of current therapies are the main objectives of healthcare professionals aiming to improve treatment outcomes for PDAC patients. However, there are still pitfalls that need to be addressed [5,6,7,8] (Figure 1):

(1) Upon diagnosis, 80% of PDAC patients find themselves in an irreversible state due to local advanced or metastatic disease. This stems from the absence of specific symptoms in PDAC patients and the unavailability of dependable early diagnostic tools.

(2) Among the 20% of patients with localized disease, surgery stands as the sole prospect for a potential cure. Nevertheless, the recurrence rate postsurgery is high (80%), raising questions about the degree of tumor removal and the inability to detect residual disease or hidden metastases during the procedure.

(3) Adjuvant therapy with FOLFIRINOX (a regimen combining 5-fluoracil, leucovorin, irinotecan, and oxaliplatin) is advised to improve surgical outcomes. Unfortunately, 60% of patients undergoing this therapy experience relapse within three years. This raises concerns about possible treatment toxicity and highlights the need to identify patients most likely to benefit from such treatments. Additionally, it is challenging to monitor the disease in real-time to evaluate the effectiveness of treatment.

(4) Neoadjuvant chemotherapy (NAC) has shown promising results in improving surgical and survival outcomes for patients with borderline resectable PDAC. However, its effectiveness in these cases remains a topic of debate, underscoring the need to identify patients who would most benefit from this approach.

Serum marker carbohydrate-antigen 19-9 (CA19-9) and contrast-enhanced computed tomography (CT) are the primary methods used to diagnose and monitor recurrence. However, CA19-9 exhibits limitations in both sensitivity (~80%) and specificity (~75%) and is not exclusive to PDAC. On the other hand, CT scans mainly detect macroscopic disease recurrence and provide limited applicability. In addition, the use of solid biopsy for histological analysis comes with reported limitations and risks [3,4,5,6,7,8,10,11,12,13,14,15].

The ideal tumor biomarker should be disease-specific, highly sensitive, and have a strong positive predictive value, especially for early detection and personalized treatments. Moreover, it should also be easily collectible and cost-effective [3,4,5,6,7,8,10,11]. Recent advancements in the understanding of PDAC biology paved the way for new opportunities in early detection and treatment methods. Liquid biopsy (LB) is an emerging minimally invasive technique that relies on the analysis of various body fluids to detect tumor biomarkers. These biomarkers are circulating tumor cells (CTCs), circulating tumor DNA (ctDNA, extracellular vesicles (EVs), microRNAs (miRNA), peripheral blood circulating RNA, tumor-educated blood platelets (TEPs), and circulating tumor vascular endothelial cells (CTECs) (Figure 1). Through the identification of multiple biomarkers, LB enables disease assessment and characterization, providing insights beyond diagnosis to include essential tumor characteristics such as cancer type and the presence of major genetic mutations [3,4,5,6,7,8,10,11,12,13,14].

Authored by clinicians, this review focuses on LB from a clinical perspective, considering it as a potentially revolutionary tool for early detection, prognosis, and residual disease monitoring, along with treatment follow-up in PDAC. The capacity of LB to dynamically track genetic changes in real-time during treatment not only holds great promise but also has the potential to significantly reshape the management of PDAC.

## 2. Methodologies and Technological Approaches for LB

### 2.1. ctDNA

In 1948, Mandel and Metais reported evidence of cell-free nucleic acid fragments in human blood [15]. Remarkably, as early as 1977, León and colleagues made intriguing statements about circulating DNA in cancer patients [13]. Extracellular DNA, also known as circulating free DNA (cfDNA), encompasses nuclear and/or mitochondrial DNA released from cells and is found in various physiological circulating fluids. The cfDNA released by tumor cells is commonly referred to as ctDNA and serves as a highly specific marker for cancer [14,15]. Studies have revealed that ctDNA often carries oncogenic mutations commonly observed in PDAC tissues, involving genes such as Kristen rat sarcoma (KRAS), cyclin-dependent kinase inhibitor 2A (CDKN2A), tumor protein 53 (TP53), and SMAD family member4 (SMAD4)/Delete in Pancreatic Cancer-4 (DPC4). Notably, mutations in KRAS or inactivation of p53 are observed in over 90% and 73% of PDAC cases with detectable ctDNA, respectively [3,8].

Various techniques, such as allele-specific polymerase chain reaction (PCR), digital PCR (dPCR), droplet digital PCR (ddPCR), beads-emulsion-amplification-magnetics (BEAMing), and next-generation sequencing (NGS), can be employed to detect ctDNA from LB samples. While the detection of ctDNA may pose challenges, combining various techniques can enhance the accuracy and efficiency of this process [3,4,5,6,7,8] (Table 1).

### 2.2. CTCs

In 1869, Ashworth reported the presence of circulating cells in peripheral blood resembling those found in a tumor from a patient with metastatic cancer [16]. Despite this discovery, CTCs were largely overlooked for over a century until recent times when they regained attention from scientists exploring their clinical applications [16]. CTCs are malignant cells that have undergone epithelial–mesenchymal transition (EMT) in the primary tumor. These cells circulate throughout the body and infiltrate lymphatic and blood vessels, potentially leading to metastasis. EMT is a crucial process in cancer initiation and progression. Initially, retaining characteristics of the epithelial cells they originated from, CTCs undergo various phenotypic changes as EMT progresses, evading apoptosis and promoting metastasis. Notably, molecules such as epithelial cell adhesion (EpCAM) and cytokeratin (CK) expressed on CTCs can alter during EMT, enhancing their migration and invasion abilities [3,4,5,6,7,8].

The extraction of CTCs from millions of blood cells is significantly challenging, as it is imperative to prevent any damage or loss to the CTCs throughout the isolation process. [3,4,5,6,7,8,10]. To tackle this challenge, various technologies have been utilized, including immunoaffinity methods that target specific antigens on the surface of tumor cells, microfluidic capture devices, and size-based separation methods. Immunoaffinity methods encompass positive enrichment through epithelial cell markers such as EpCAM or CK, as well as negative enrichment through CD45 to eliminate leukocytes [3,4,5,6,7,8,10]. Physical property-based methods include density gradient centrifugation based on size-exclusion chromatography. At times, a synergistic approach employing techniques based on the physical and biological properties of CTCs is utilized for their isolation (Figure 1 and Table 1).

### 2.3. EVs

EVs are secreted by cells and play a pivotal role in intercellular communication, harboring a diverse array of proteins, DNA, miRNAs, and lipids on their surface or within them. Recent studies highlight the ability of EVs to modulate the tumor immune microenvironment and contribute to PDAC progression through the induction of cell migration and enhancement of tumor aggressiveness. Interestingly, both PDAC cells and stroma cells have been observed to release and uptake EVs. Notably, a recent study demonstrated that carcinoma-associated fibroblast-derived EVs could induce cell migration and EMT in PDAC cells through CD9, thereby enhancing tumor aggressivity [3,4,5,6,7,8,10] (Figure 1 and Table 1).

Various methods are available to capture and isolate circulating EVs. These include size-based, density-based, and affinity-based techniques, each offering different levels of sensitivity and specificity [3,4,5,6,7,8,10]. Investigating the RNA content of EVs revealed its prognostic value for overall survival (OS) and demonstrated that early alterations in EVs could function as indicators of disease relapse [8,10].

## 3. Impact of LB in Early and Differential Diagnosis of PDAC

The challenge in the clinical practice in this pathology (pancreatic intraepithelial neoplasia (PanIN) or early stage PDAC) lies in the early detection to improve the prognosis of PDAC. LB holds the potential to enhance detection without subjecting the patient to the risks associated with aggressive diagnostic tools [5]. A systematic review and meta-analysis from Zhu and colleagues described the studies performed on LB methods in detecting PDAC [3]. The sensitivity and specificity of LB were 80% and 89%, respectively, with an area under the curve (AUC) of 0.936 [3,5] (Figure 1 and Table 1).

The analysis of various body fluids, including pancreatic juice, saliva and urine, has been investigated for the early detection of PDAC. Mutations in KRAS have been detected in the pancreatic juice and in biliary cytobrush specimens, where selective detection has been observed in PDAC patients over benign lesions [17]. Mutation levels of KRAS2 were higher in patients with PDAC compared to chronic pancreatitis (CP). In CP, telomerase activity and alterations in genes such as CDKN2A/p16, TP53 and SMAD4/DPC4 could aid in PDAC diagnosis [18,19]. The analysis of biomarkers in saliva offers an easy and noninvasive way to search for mutations in RNAs and miRNAs of genes such as MBD3L2, KRAS, ACRV1 and DPMI, among others, which could assist in differentiating pancreatic cancer patients from CP patients and healthy control [20]. Other biomarkers in RNAs and miRNAs have been described [21,22]. Analysis of genetic mutations in urine could also facilitate detection of PDAC [23,24]. 

Importantly, the efficacy of LB to detect PDAC varies depending on the tumor stage and biomarker type. For example, ctDNA is less frequently detected in patients with resectable disease compared to those with unresectable disease [5]. Here, we outline the utility of various biomarkers in detecting PDAC within the general population and among specific risk groups (Figure 1 and Table 1).

### 3.1. ctDNA

Some studies found that cfDNA is detected at higher levels from PDAC patients compared to pancreatic neuroendocrine tumors or CP patient [25] and have been associated with poor disease-specific survival [25,26,27]. Of all mutated genes detected from ctDNA in PDAC patients, KRAS is the one most frequently found (50–90%). Although mutations can also be found in healthy controls and patients with CP, its mutation levels are significantly higher in PDAC [7,26]. 

In their review, Zhu and colleagues emphasized that although the sensitivity of ctDNA is marginally lower than that of CTCs, ctDNA provides considerably higher specificity [3]. Notably, while the detection of ctDNA is deemed appropriate for the diagnosis of PDAC, it is not considered suitable for screening purposes. The limited sensitivity of ctDNA in early-stage PDAC is attributed to minimal cellular necrosis at this stage, resulting in the release of only a small quantity of ctDNA into the peripheral bloodstream [3].

### 3.2. CTCs

Pancreatic cells can indeed be identified in the bloodstream even before tumor development. In contrast to healthy individuals, these cells are detectable in 33% of patients with cystic lesions [28] and in 73% of those with PDAC [28,29]. Furthermore, other studies have noted the presence of CTCs in varying proportions in benign, premalignant, or malignant lesions, but not in healthy controls [30].

The diagnostic efficacy of CTCs, however, has been a topic of debate among researchers due to their inconsistent sensitivity, which ranges from 21% to 100% [31]. In a study by Ankeni and colleagues, a NanoVelcro CTCs microfluidic chip was employed to analyze 100 sequential samples from pretreatment PDAC patients. The results showed CTCs in 54 out of 72 confirmed PDAC patients, demonstrating a sensitivity of 75% and a specificity of 96.4% when used for diagnostic purposes [32].

Given the technical variability in detecting CTCs, some researchers recommend combining the detection of CTCs with other biomarkers to enhance sensitivity up to 100% and specificity to 80% [33]. Although the specificity of CTCs is lower than that of EVs, their utilization still presents considerable diagnostic potential in PDAC detection [3].

### 3.3. EVs

EVs from pancreatic cells are easily detectable in peripheral blood owing to their abundant levels, a consequence of the exocrine function of these cells, and possess a longer half-life compared to ctDNA. Interestingly, KRAS mutations in ctDNA were found in 7.4% and 14.8% of healthy donors when tested from exosomal DNA (exoDNA) or cfDNA, respectively [34], and in 13% of CP patients [35]. Research by Melo et al. demonstrated that glypican-1 (GPC1) levels in EVs were significantly higher in PDAC patients compared to those with benign pancreatic diseases or healthy individuals, showcasing remarkable sensitivity and specificity (100%) [36]. Moreover, studies by Zhang et al. [37] and Lewis et al. [38] employed ‘chips’ to detect GPC1, aiming to enhance sensitivity and specificity in distinguishing PDAC from healthy samples. Other studies have shown that combining EVs analysis with CA19-9 could improve specificity [39]. Notably, EVs analysis proves useful for both diagnosing and screening PDAC, offering superior diagnostic value compared to other techniques, partly due to its high AUC of 0.9819 [3]. To summarize, among various detection methods, EVs exhibit the highest diagnostic efficacy, sensitivity, and AUC [3].

### 3.4. miRNAs

miRNAs are noncoding, single-stranded RNA molecules up to 22 nucleotides long that function as posttranscriptional gene expression regulators [40]. Various miRNAs have been utilized to differentiate PDAC patients from those with benign lesions, CP, and healthy controls, and a JAMA-published study reported 38 significantly dysregulated miRNAs in PDAC patients compared to controls [41]. These include miR-223, miR-23b-3p, miR-100, miR-205, miR-192-5p, a six-miRNA panel, miR-483-3p, miR-99 (a and b), miR-21, miR-25, and miR-205, among others. The detection of diverse miRNAs could enhance diagnostic accuracy and distinguish PDAC from healthy individuals [42]. In pancreatic conditions such as intraductal papillary mucinous neoplasm [43] or pancreatic intraepithelial neoplasia [44], miRNAs have been instrumental in differentiating patients with high-grade dysplasia or early-stage PDAC from healthy controls. Importantly, a meta-analysis revealed that miRNAs offer a sensitivity of 79% and specificity of 74% for early PDAC diagnosis. Similar to other biomarkers, combining analysis of miRNAs with CA19-9 analysis improved the quality of data, yielding an AUC of 0.84 [45]. Moreover, whilst miRNA analysis from pancreatic juice demonstrated a specificity of 88% and sensitivity of 87%, the inclusion of serum CA19-9 levels enhanced sensitivity to 91% and specificity to 100% [46]. In saliva samples, hsa-miR-21, hsa-miR-23a, hsa-miR-23b, and miR-29c were significantly upregulated in PDAC patients compared to controls, reporting sensitivities of 71.4%, 85.7%, 85.7%, and 57%, respectively, and a specificity of 100% [47].

Numerous studies exploring biomarker combinations in healthy individuals, CP, and PDAC patients [48,49] have produced various results, with a common trend pointing towards higher specificity when combining biomarkers. For instance, one study demonstrated that using at least two biomarkers among CA19-9, CTCs, or ctDNA achieved a sensitivity and specificity of approximately 80% and 90%, respectively [50,51]. Another study confirmed that combining CTCs with CA19-9 elevated the positive diagnostic rate for PDAC to 97.5% [52].

## 4. Role of LB after Resection of PDAC

Surgery stands as the singular potentially curative intervention for PDAC, given the tumor’s notable resistance to chemotherapy, radiation, and immunotherapy. Attaining an R0 resection represents the optimal opportunity for patients to achieve a favorable prognosis, resulting in an enhanced 5-year survival rate ranging from 8% to 25% [3,4,5,6,7,8,10,11]. Despite undergoing curative resection, disease recurrence significantly impacts postoperative outcomes, with over 70% of resected PDAC patients succumbing to recurrent disease [3,4,5,6,7,8]. Consequently, there is a critical need for an effective strategy to identify minimal residual disease (R1) during or postsurgery and to anticipate the risk of recurrence [11]. LB emerges as a promising approach to monitor disease progression in PDAC following surgical intervention [3,4,5,6,7,8,10,11] (Figure 1 and Table 1).

### 4.1. ctDNA

Despite the initial hypothesis suggesting that tumor components are released into the bloodstream following manipulation, a recent meta-analysis revealed that surgical resection of resectable primary tumors plays a role in the negativization of ctDNA [11]. However, this comprehensive analysis failed to establish a significant impact of ctDNA negativization on the overall survival (OS) and disease-free survival (DFS) rates of PDAC patients. The authors emphasized the considerable heterogeneity observed within the analyzed studies, attributing it to variations in postoperative determinations of ctDNA, which extended beyond 24 h after the removal of surgical specimens. Considering the relatively short half-life of ctDNA, ranging from minutes to hours, delayed determinations increase the likelihood of encountering false-negative results. On the other hand, the elevation in ctDNA levels induced by surgical trauma can persist for 2 to 4 weeks, potentially concealing persistent ctDNA in patients experiencing relapse. Consequently, the authors recommended conducting LB between 2 and 4 weeks postsurgery to minimize the risk of false negatives [3,4,5,6,7,8].

Recent studies have indicated that the detection of ctDNA in preoperative blood samples from PDAC patients may identify candidates for NAC. Additionally, elevated ctDNA levels were associated with an increased risk of death, and patients with a LB after surgery showing as positive for ctDNA exhibited a higher recurrence rate [11]. The persistence of ctDNA positivity postsurgical resection is predictive of a shorter DFS, possibly indicating the presence of occult micrometastases or residual local disease, supporting the consideration of additional adjuvant treatment [7,8,10,11].

### 4.2. CTCs

In a recent randomized trial, the authors implemented a no-touch pancreaticoduodenectomy, involving the manipulation of the tumor only after the complete isolation of vascular and lymphatic drainage vessels [12]. The study reported a reduction in CTCs in the portal vein when employing the no-touch technique but no discernible improvement in OS compared to standard surgery. In contrast, a meta-analysis conducted by Vidal and colleagues failed to demonstrate a decreased release of tumoral components after surgical manipulation employing the no-touch technique, and no superior OS and DFS were observed compared to standard surgery [11].

Importantly, the presence of CTCs has been shown to progressively increase in advanced diseases. For instance, a study revealed that patients with occult metastases of PDAC exhibited significantly higher CTCs counts than those without metastatic disease. Therefore, considering CTC levels becomes crucial in understanding the disease state and thus determining whether NAC should be favored over surgical treatment [10]. Additionally, a prospective study assessed CTC levels in preoperative PDAC patients, establishing a direct correlation between elevated CTC levels and disease recurrence at one year in patients undergoing resection [8]. 

### 4.3. EVs

Several studies have suggested that EVs derived from tumors can serve as indicators of the response to surgery and treatment, offering a potential avenue for a reliable marker in PDAC. Similar to other markers, the levels of miRNAs originating from EVs return to normal within 24 h of PDAC resection. However, persistently elevated levels postsurgery may indicate the presence of hidden metastasis. Such cases warrant vigilant follow-up, and individuals may require additional treatment following surgery [10]. A study investigating miR-451a, miR-4525, and miR-21 from EVs obtained from portal venous blood during pancreatectomy revealed elevated levels emerging as an independent prognostic factor for OS and DFS [8].

## 5. Assessment of NAC in PDAC

Evidence from a multicenter, randomized, intention-to-treat phase III trial suggests that NAC enhances long-term OS in resectable and borderline-resectable PDAC [53]. Concerns among clinicians and surgeons regarding delays in surgical treatment due to NAC have persisted particularly for patients who experience disease progression under NAC. In the context of PDAC, careful consideration should be given to interpreting the clinicopathological response using radiological criteria alone (RECIST). Due to the fibrosis and specific changes in the extracellular matrix that can occur after NAC, tumor size may not show significant alterations. Thus, relying solely on this method for evaluation may not provide an adequate correlation with these events occurring in the tumor bed after NAC. Therefore, in practice, therapeutic decisions are typically grounded on the nonprogression of the disease under chemotherapy rather than on a positive local response.

Advancements in the molecular investigation of PDAC have given rise to a classification system comprising four types based on 32 recurrently mutated genes organized into 10 pathways [54]. Beyond identifying potential new therapeutic targets, the molecular disorders utilized to determine these PDAC subtypes could also prove valuable in the prognosis and monitoring of NAC in PDAC. While LB is a novel tool and its systematic use in the follow-up of NAC has not been extensively scrutinized, there are some promising results that merit consideration. Although the evolution of biomarkers used in LB in patients with resectable tumors during NAC has not been fully monitored, studies focusing on disease progression during treatment in these patients consistently indicate a poor prognosis when these markers persist throughout the disease course.

Serial assessment of mutated KRAS ctDNA in advanced PDAC patients has revealed a kinetic pattern correlating with radiological response, progression-free survival, and OS. Importantly, low levels of mutated KRAS ctDNA during therapy serve as an early indicator of positive treatment response [55]. A prospective longitudinal study demonstrated significantly lower CTCs in patients receiving NAC compared to those undergoing initial resection. The authors proposed a risk assessment score based on the difference in CTC levels, accurately predicting disease recurrence within the next two months, with 75% and 84% accuracy for chemotherapy-naïve or post-NAC patients, respectively [56]. According to Yin and colleagues, even PDAC patients who showed a complete response to NAC still had somatic CTCs and ctDNA mutations present. This suggests potential early recurrence and reduced survival rates. The authors proposed a novel approach for assessing pathological response in PDAC, combining genomic analysis of resected specimens with LB data [57]. This concept, termed molecular complete response, could become the future gold standard for evaluating initial treatment response (Figure 1 and Table 1).

## 6. The Use of LB and in Recurrence and Prognosis of PDAC

The application of LB has yielded promising results in determining the prognosis and predicting the likelihood of disease recurrence (Figure 1 and Table 1).

### 6.1. ctDNA

Mendel and colleagues reported that ctDNA levels not only rise in cancer patients but are significantly higher in those with metastatic disease compared to nonmetastatic cases. Persistently high or rising ctDNA levels potentially indicate a lack of response to treatment and/or relapse and serve as a poor prognostic indicator. Serum DNA proves valuable for therapy evaluation and regimen comparison, although its diagnostic utility is limited given that a relatively high percentage of cancer patients present apparently normal levels [13].

A study reported that mutations in codon 12 of the KRAS gene lacked associations with clinicopathological parameters such as vascular encasement, tumor mass, lymphatic invasion, and metastasis [58]. However, elevated cfDNA levels in plasma (>62 ng/mL) were significantly associated to lower OS, vascular encasement, and metastasis in PDAC. Another study reported ctDNA as an independent prognostic biomarker for OS in advanced disease, with ctDNA-positive patients exhibiting significantly shorter OS (6.5 months) compared to ctDNA-negative counterparts (19.0 months) [59]. Interestingly, mutations in KRAS, TP53, SMAD4, CDKN2A, and TGFBR2 were detected in individuals with PDAC. Moreover, it was reported that preoperative ctDNA-positive patients have notably poorer OS compared to ctDNA-negative patients, which is only observed in those who undergo pancreatic resection [60]. Further studies showed that detection of ctDNA before or after the initiation of chemotherapy correlated with shorter PFS and OS [61]. Finally, Nakano and colleagues identified postoperative serum KRAS mutations as an independent prognostic factor for DFS in PDAC patients undergoing curative pancreatectomy. The shift from preoperative wild-type KRAS to postoperative mutant KRAS was also an independent prognostic factor for OS [62].

In a study comparing KRAS ctDNA with CA19-9 over time, it was observed that presurgery or prechemotherapy KRAS-mutated ctDNA had no clear association with prognosis and recurrence, and increased CA19-9 levels were significantly associated with recurrence but not prognosis [63]. The detection of KRAS ctDNA, was significantly associated with prognosis irrespective of recurrence and served as a predictive factor for prognosis of nonsurgical patients. Importantly, patients showing no emergence of KRAS ctDNA within a year after surgery displayed a significantly better prognosis, irrespective of recurrence. Moreover, regardless of tumor resection, the detection of KRAS ctDNA emerged as the sole independent prognostic factor, and the absence or disappearance of KRAS KRAS ctDNA within six months of treatment correlated with therapeutic responses to first-line chemotherapy. These findings highlight the critical role of KRAS in predicting therapeutic responses and improving patient outcomes [63].

### 6.2. CTCs

A study reported that individuals with detectable CTCs experienced a markedly reduced OS duration of 88 days compared to those without CTCs, who had an OS of 393 days [64]. Unfortunately, the study also underscored the limitations of detecting CTCs using the CellSearch^®^ system, as it has a low detection capability, primarily identifying CTCs in patients with metastatic disease where treatment options are limited. Another study demonstrated lower survival rates over a follow-up period of one-and-a-half years in patients suffering from PDAC [65]. Interestingly, a threshold of ≥3 CTCs in 4 mL of venous blood proved sufficient to distinguish between local/regional and metastatic disease, linking the latter to a poorer prognosis in patients with distant disease [32].

The detection or absence of CTCs has surfaced as an independent prognostic factor in OS, alongside the presence of peritoneal disease and liver metastases. An analysis examining the correlation between clinical characteristics and CTC status revealed a statistically significant association with the occurrence of liver metastases but not with peritoneal dissemination. These findings suggest that leveraging CTCs as a prognostic biomarker could offer benefits in the management of PDAC patients [66].

### 6.3. EVs

In individuals with pancreatic tumors, a high expression of macrophage migration inhibitory factor (MIF) was detected from PDAC-derived EVs, particularly in patients who subsequently developed liver metastasis, suggesting MIF as a potential prognostic indicator for the development of PDAC liver metastasis. In another study, miR-222 contained within EVs derived from PDAC cells was identified as an independent risk factor for patient survival [67]. GPC1, which is present in PDAC cells and neighboring stromal fibroblasts, is also noteworthy. While circulating GPC1 EVs may not definitively distinguishing PDAC from benign pancreatic conditions, elevated levels were significantly found in larger tumors (>4 cm) [68]. Hypoxia-induced programmed cell death activates p53-independent apoptotic pathways, leading to local invasive growth, perifocal tumor cell dissemination, and regional and distant tumor cell metastasis, and a poor prognosis [69]. Additionally, pancreatic cancer cells produce miR-301a-3p in a hypoxic environment, which is also enriched in EVs, predicting a late TNM stage and reduced survival [70].

## 7. Conclusions

LB heralds a transformative era in the early detection, diagnosis, and management of PDAC. Improving survival rates necessitates the development of biomarkers that are both highly sensitive and specific and capable of diagnosing PDAC in its early stages. However, challenges such as nonstandardized detection techniques, high costs, limited accessibility, and the absence of clear cut-off levels currently hinder the widespread clinical adoption of circulating biomarkers. LB emerges as a noninvasive method to assess PDAC treatment responses, enabling medical teams to promptly adapt treatment plans or switch to more suitable alternatives as the disease progresses. While LB might not suffice as a standalone evaluation tool, its potential to complement existing methodologies promises a more comprehensive and precise assessment. Although research indicates LB’s ability to predict survival outcomes in PDAC, more extensive studies are required to fully comprehend its clinical implications. Anticipated advances in research dedicated to identifying, refining, and clinically validating biomarkers will further enhance LB’s utility. Ultimately, this noninvasive approach is poised to establish a new standard in PDAC diagnosis, treatment monitoring, prognostication, and follow-up, potentially becoming a recommended practice in primary pancreatic cancer treatment guidelines.

## Figures and Tables

**Figure 1 ijms-25-01640-f001:**
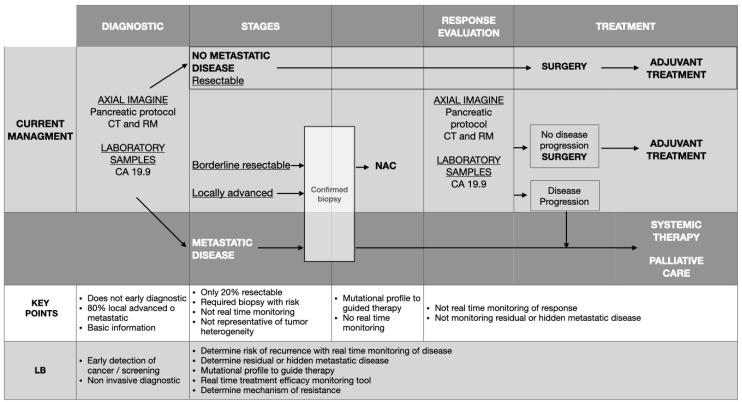
Clinical application of liquid biopsy in pancreatic ductal adenocarcinoma [9]. LB: Liquid biopsy; NAC: Neoadjuvant chemotherapy.

**Table 1 ijms-25-01640-t001:** Comparison between biomarkers of pancreatic ductal adenocarcinoma [4,6,8].

	ctDNA	CTCs	EVs
Target	KRAS, TP53, CDKN2A, SMAD4, BRAF, PIK3CA, ADAMTS1, BNC1, 5MC, H2AZ, H2A1.1, H3K4me2, h2ak119ub	CD45, CEP8, CK, EpCAM	KRAS, TP53, RNA: miRNA, longRNA Proteins markers: EFGR, EPCAM, MUC-1, GPC-1, WNT2
Isolation	Blood	Blood	Body fluids
Tumor information	Epigenetic information	DNA, RNA, Protein	DNA, RNA, Protein
Technological approaches	qPCR, dPCR, ddPCR, NGS, commercial kits	Immunoaffinity, Physical methods (size and density)	Density-based, size-based, affinity-based, commercial kits
Advantages	qPCR: Fast and low-cost dPCR: High sensitivity/Specificity NGS: capability to screen for a broad range of genetic variants using high DNA input	Immunoaffinity: Specific, label-free obtained Physical methods: Fast, simple, Low-cost, label-free obtained	Density-based: low cost. Independent of marker expression. Size-based: Low-cost, fast, Independent of marker expression. Affinity-based: Specificity. High purity. Commercial kits: Simple, fast.
Disadvantages	General: No early stages qPCR: Low sensitivity. Only points mutations. dPCR: High cost. Only points mutations. NGS: Variable sensitivity. High cost.	General: Isolation complex and expensive. Technical variability Immunoaffinity: capture only one subpopulation. Low purity. Physical methods: Needs immuno-labeling techniques to distinguish CTCs	General: Isolation complex by contamination and expensive Density-based: Time, high volume sample, can damage EVs. Size based: contamination. Affinity-based: low sample yield. Commercial kits: High cost.
Sensitivity (S) (%)	34–71% KRAS mutations: codons 12, 13, 61, in different stages.	73–76% CD45/CEP8 100% Mt, 58% resectable Anti-EpCAM portal vein Blood	67% ES, 80% LA, 85% Mt KRAS mutations in exoDNA 50% ES GPC1 miRNAs Increased expression
Specificity (Sp) (%)	75–81% Mutations KRAS exon 2	68% CD45/CEP8	90% ES GPC1
Combined techniques (%)	S: 85–98%, Sp: 77–81% ctDNA (KRAS exon 2) with CA19.9 S: 47% ctDNA (KRAS MAFs) with CA19.9	S: 100%, Sp: 80% CTCs.with EVs	NR
Application	No suitable for screening of PDAC Monitoring postoperative minimal residual disease Predictor of disease recurrence and prognosis	Not present in healthy controls Variable sensitivity in early diagnosis Excellent specificity. Follow-up of disease recurrence and prognosis Functional analysis drug resistance	The highest sensitivity and specificity in early detection Evaluated response of resection or any therapy Biotherapeutic application

BEAMing: beads-emulsion-amplification-magnetics; CEP8: chromosome 8 centromere; ctDNA: circulating tumor DNA; CTC: circulating tumor cells; dPCR: digital polymerase chain reaction; ddPCR: droplet digital PCR; EpCAM: epithelial cell adhesion molecule; ES: early stages; EVs: extracellular vesicles; GPC1: glypican-1; LA: locally advanced; miRNA: micro-RNA; Mt: metastatic; NGS: next-generation sequencing; NR: not reported.

## Data Availability

No new data were created or analyzed in this study. Data sharing is not applicable to this article.

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
