# Peer review of "Clinical Application of Liquid Biopsy in Pancreatic Cancer: A Narrative Review"

_ijms, 2024, doi:10.3390/ijms25031640_

Round 1

Reviewer 1 Report

Comments and Suggestions for Authors

The article discusses the importance of identifying new markers for pancreatic ductal adenocarcinoma and the potential of liquid biopsy as a non-invasive and real-time monitoring technique for early diagnosis and personalized treatment. The text aims to review the progress and potential clinical applications of liquid biopsy as a technique for detecting pancreatic cancer.  

This review has a sufficient level of scientific information. However, there are important flaws in the manuscript listed below:

- The title does not match the content and could be changed to better match the text. The text is more focused on diagnostic and prognosis rather than treatment monitoring.

- In this review paper, only one simple table is mentioned. I recommend including more tables and possibly figures to better explain the content to the readers.

- Table 1 is not clear. What does the " All of them" represent in the application section? This table needs to be completed and explained further, including the advantages, sensitivity and specificity of each method and ….

- As there have been previous reviews on biomarkers of pancreatic cancer, it is important to highlight the novelty of this review.

- More explanations about molecular diagnosis are recommended to be included in the review.

Comments on the Quality of English Language

Minor editing of English language required

Author Response

AUTHOR RESPONSE TO REVIEWERS

Response: We would like to thank the reviewers for their valuable comments and the recommendations received and for their constructive nature, with which our manuscript has been improved. We have carried out a review of the formal aspects, as well as the wording of the text, and corrected the errors pointed out by the reviewers. Below, we provide thorough responses to each comments made by the reviewers.

Reviewer: 

The article discusses the importance of identifying new markers for pancreatic ductal adenocarcinoma and the potential of liquid biopsy as a non-invasive and real-time monitoring technique for early diagnosis and personalized treatment. The text aims to review the progress and potential clinical applications of liquid biopsy as a technique for detecting pancreatic cancer.  

This review has a sufficient level of scientific information. However, there are important flaws in the manuscript listed below: 

  1. The title does not match the content and could be changed to better match the text. The text is more focused on diagnostic and prognosis rather than treatment monitoring.
  • Response: As suggested by the reviewer, the relevant modifications have been made to the manuscript title. 
  • Location: Title 

  1. In this review paper, only one simple table is mentioned. I recommend including more tables and possibly figures to better explain the content to the readers.
  • Response: As suggested by the reviewer, a table is included and more information is added to the previous table. 
  • Location: Table 1 and 2

  1. is not clear. What does the " All of them" represent in the application section? This table needs to be completed and explained further, including the advantages, sensitivity and specificity of each method and ….
  • Response: the table has been modified (now table 2), as suggested by the reviewer. 
  • Location: Table 2

  1. As there have been previous reviews on biomarkers of pancreatic cancer, it is important to highlight the novelty of this review.
  • Response: According to the valuable suggestion of the reviewer, has been included the novelty of this review. 
  • Location: Introduction

  1. More explanations about molecular diagnosis are recommended to be included in the review
  • Response: In accordance with the reviewer's helpful suggestion, More information about molecular diagnosis has been included. 
  • Location: “Impact of liquid biopsy in early diagnoses/detection and differential diagnoses in PDAC” chapter and “Methodologies and technological approaches for LB” chapter

Reviewer 2 Report

Comments and Suggestions for Authors

The manuscript focuses on a systemic revision of literature data about the role of liquid biopsy in the clinical management of PDAC patients represents a technically correct and timely relevant manuscript able to elucidate how liquid biopsy may impact on the stratification of PDAC patients.

-. In the introduction section, please, could the authors evaluate the opening challenges about the clinical management of PDAC patients?

- In the text, please, could the authors substitute exosomes with "extracellular vescicles"? In my opinion, exosomes is not correct and could be replaced with a more comprehensive and inclusive term related to this biomarker

- In the text, please, could the authors also add a brief description of the most suitable technological approaches available to evaluate aforementioned analytes?

- In the table 1, please, could the authors also add how each analyte may be evaluated?

Comments on the Quality of English Language

Moderate editing of English language required

Author Response

AUTHOR RESPONSE TO REVIEWERS

Response: We would like to thank the reviewers for their valuable comments and the recommendations received and for their constructive nature, with which our manuscript has been improved. We have carried out a review of the formal aspects, as well as the wording of the text, and corrected the errors pointed out by the reviewers. Below, we provide thorough responses to each comments made by the reviewers.

Reviewer: 

The manuscript focuses on a systemic revision of literature data about the role of liquid biopsy in the clinical management of PDAC patients represents a technically correct and timely relevant manuscript able to elucidate how liquid biopsy may impact on the stratification of PDAC patients 

  1. In the introduction section, please, could the authors evaluate the opening challenges about the clinical management of PDAC patients
  • Response: According to the valuable suggestion of the reviewer, has been included the this information and table 1 with this subject. 
  • Location: Introduction and table 1

  1. In the text, please, could the authors substitute exosomes with "extracellular vescicles"? In my opinion, exosomes is not correct and could be replaced with a more comprehensive and inclusive term related to this biomarker
  • Response: In accordance with the point raised by the reviewer, this term has been changed. 
  • Location: Entire manuscript

  1. In the text, please, could the authors also add a brief description of the most suitable technological approaches available to evaluate aforementioned analytes?
  • Response: The name of the chapter has been changed to “Methodologies and technological approaches for LB” because we think that this chapter describes what the reviewer requests. But if the reviewer considers that we should add more information or modify it, it will be done. 
  • Location: “Methodologies and technological approaches for LB” chapter

  1. In the table 1, please, could the authors also add how each analyte may be evaluated.
  • Response: In accordance with the reviewer's helpful suggestion, more information has been included. Table 1 has been renamed as table 2.
  • Location: Table 2

Round 2

Reviewer 1 Report

Comments and Suggestions for Authors

-